# Application of Indocyanine Green Fluorescence Imaging for Tumor Localization during Robot-Assisted Hepatectomy

**DOI:** 10.3390/cancers15174205

**Published:** 2023-08-22

**Authors:** Masahiko Kinoshita, Takahito Kawaguchi, Shogo Tanaka, Kenjiro Kimura, Hiroji Shinkawa, Go Ohira, Kohei Nishio, Ryota Tanaka, Shigeaki Kurihara, Shuhei Kushiyama, Takeaki Ishizawa

**Affiliations:** 1Department of Hepato-Biliary-Pancreatic Surgery, Osaka Metropolitan University Graduate School of Medicine, Osaka 545-8585, Japan; pikopiko0128@yahoo.co.jp (M.K.); takahitokawaguchi@gmail.com (T.K.); kenjiro@omu.ac.jp (K.K.); hirojishinkawa9876@gmail.com (H.S.); m1153@omu.ac.jp (G.O.); m1155123@omu.ac.jp (K.N.); taanaakaa3364@gmail.com (R.T.); s-kurihara0731@hotmail.com (S.K.); bbc08ks@gmail.com (S.K.); 2Department of Hepato-Biliary-Pancreatic Surgery, Izumi City General Hospital, Izumi City 594-0073, Japan; shogotanaka@omu.ac.jp

**Keywords:** indocyanine green, fluorescence imaging, robot-assisted hepatectomy, hepatocellular carcinoma, intrahepatic cholangiocarcinoma

## Abstract

**Simple Summary:**

While indocyanine green (ICG) fluorescence imaging has widely been used as an intraoperative navigation tool, its efficacy for visualization of hepatic tumors remains to be clarified, especially in robot-assisted hepatectomy (RAH). In our present study, fluorescence imaging identified tumors on hepatic surfaces before hepatic transection in 26/31 tumors. In eight tumors, fluorescence signals were detected from hepatic raw surfaces during parenchymal dissection, enabling surgeons to adjust transection planes to determine surgical margins. As a result, pathological examinations found negative surgical margins at the site of dissected hepatic parenchyma in all tumors identified using fluorescence imaging. On the contrary, a positive surgical margin surrounding dissected hepatic parenchyma was observed in one of two patients in whom ICG was contraindicated. ICG fluorescence imaging enables the identification of hepatic tumors easily even in the setting of RAH, which may be useful for determining surgical margins.

**Abstract:**

The efficacy of indocyanine green (ICG) fluorescence imaging for visualizing hepatic tumors in robot-assisted hepatectomy (RAH) should be validated. This study included 30 consecutive patients with 33 collective tumors who underwent RAH. ICG was administered at a dose of 0.5 mg/kg before surgery. ICG fluorescence imaging was performed intraoperatively. In total, 28 patients with a combined total of 31 tumors underwent ICG fluorescence imaging. Further, 26 (84%) tumors were identified on hepatic surfaces prior to hepatic transection. The fluorescence signals of eight tumors were detected on hepatic raw surfaces during parenchymal dissection, thereby enabling surgeons to adjust the transection planes to ensure appropriate surgical margins. One patient with intrahepatic cholangiocarcinoma tested positive for cancer cells at the dissected stump of the bile duct. However, in all patients in whom ICG fluorescence imaging was used, negative surgical margins were achieved at the site of the dissected hepatic parenchyma. On the other hand, one of two patients with ICG contraindications had a positive surgical margin surrounding the dissected hepatic parenchyma. The median operative time and volume of blood loss were 259 (range: 124–594) min and 150 (range: 1–1150) mL, respectively. ICG fluorescence imaging facilitates the easy identification of hepatic tumors, even in RAH. Hence, it can be useful for confirming appropriate surgical margins.

## 1. Introduction

Indocyanine green (ICG) was initially used as a reagent for assessing hepatic function before liver surgery [1,2,3,4]. Due to the fluorescent properties of ICG, it has been widely applied in the field of hepatobiliary surgery for the intraoperative identification of hepatobiliary structures since the late 2000s [5,6,7,8]. ICG fluorescence imaging is based on the emission of fluorescence signals by protein-bound ICG if illuminated by near-infrared light [9]. In the field of hepatobiliary surgery, this technique can be used for bile duct visualization, tumor localization, and hepatic segmentation [5,6,7,10,11,12]. Previous studies have assessed the usefulness of ICG fluorescence imaging in hepatobiliary surgery [13,14,15,16,17]. In particular, due to poorer palpation and ease of use, the visualization of hepatic tumors is a more efficient technique in laparoscopic hepatectomy than in open hepatectomy [18].

Robot-assisted surgery has been utilized in multiple surgical fields [19,20,21]. Over the last decade, there has been a constant increase in the application of robot-assisted hepatectomy (RAH) [22,23]. Robotic platforms such as three-dimensional visualization and its seven degrees of freedom have advantages. Hence, they can be beneficial compared with laparoscopic surgery in more technically demanding resections [24,25,26]. Although there are limited studies comparing RAH and laparoscopic hepatectomy [27,28], RAH can broaden the applications of minimally invasive surgery [23]. However, the complete loss of palpation in robot-assisted surgery may increase surgical difficulty, including tumor detection, during RAH. The identification of tumors via ICG fluorescence imaging may, to some extent, mitigate this aspect of RAH, giving it the potential to be a highly useful tool alongside laparoscopic hepatectomy. However, there are only a few reports regarding this notion, and the number of patients who are viable to undergo the identification of tumors via ICG fluorescence imaging in RAH is limited [17,29].

This current retrospective study aimed to investigate the efficacy of ICG fluorescence imaging in identifying hepatic tumors in patients who underwent RAH.

## 2. Materials and Methods

### 2.1. Study Cohort

This study included 30 patients with a total of 33 tumors who underwent RAH at Osaka Metropolitan University Hospital between May 2022 and June 2023. In total, 23 patients were men and 7 were women. The median age of the patients was 76 (range: 52–84) years. The surgical and pathological outcomes of all patients were investigated to assess the efficacy of ICG fluorescence imaging for the retrospective visualization of tumors during RAH.

This retrospective study was performed in accordance with the ethical guidelines of the Declaration of Helsinki. It was approved by the ethics committee of Osaka Metropolitan University (no. 2022-142). All participants provided written informed consent.

### 2.2. Patient and Public Involvement

The present study did not involve any patient or public partnership.

### 2.3. ICG Fluorescence Imaging for Identifying Hepatic Tumors

Almost all patients in our department, except those with an iodine allergy, routinely underwent systemic injection of ICG (0.5 mg/kg, 2–14 days before surgery) to allow calculation of the ICG retention rate, which has been used as a routine liver function test [1,2,3,4]. The tumor or the surrounding area is often fluorescently labeled at the time of surgery due to the preoperative intravenous ICG infusion used in this test [6].

### 2.4. Standard Surgical Procedures

The da Vinci Xi system^®^ (Intuitive Surgical Inc., Sunnyvale, CA, USA) was utilized for RAH in our department. The standard patient’s position was an open-legged position. To resect the right posterior regions, the patient was placed in the left semi-lateral position with the right arm suspended. Four trocars in the da Vinci Xi system^®^ were placed in an arching line toward the target. An assistant trocar was placed at the lateral end of the right subcostal line. The surgeon lifted the liver with the TIP-UP (4th arm) and dissected the hepatic parenchyma using the clamp-crushing method with a suction irrigator (1st arm) and Bipolar Maryland^®^, Vessel sealer^®^, or Harmonic^®^ (3rd arm).

Targeted tumors were mainly identified via intraoperative contrast-enhanced ultrasound sonography. That is, perfluorobutane microbubbles appeared as hypoechoic nodules on Kupffer phase images (Sonazoid^®^; GE Healthcare, Oslo, Norway) [30]. In addition, the Firefly Mode^TM^ on the da Vinci Xi system^®^, which facilitates assessment with ICG fluorescence imaging using near-infrared illuminated light, was utilized for tumor identification. In principle, the additional resections of lesions that were newly detected via ICG fluorescence imaging were considered only if these lesions were identified via contrast-enhanced ultrasonography.

ICG fluorescence imaging was used to identify the tumor locations on hepatic phrenic surfaces. In addition, it was used during hepatic parenchymal dissection, leading to negative surgical margins. If the fluorescence signal could be observed on raw dissected surfaces during hepatic parenchymal dissection, the transection plane was adjusted with reference to the fluorescence signal, followed by re-assessment via ultrasonography.

### 2.5. Definition

Patients who tested positive for the hepatitis C virus antibody or the hepatitis B virus surface antigen were diagnosed with viral hepatitis. Patients with pathological steatosis or fibrosis in noncancerous liver tissues caused by alcohol abuse (>80 g of ethanol per day for 5 years) were diagnosed with alcoholic liver disease. A diagnosis of nonalcoholic steatohepatitis (NASH) was based on the following: (1) a history of no or limited daily alcohol intake (<20 g in women and <30 g in men), (2) the presence of hepatic steatosis confirmed via histological examination, and (3) the exclusion of other liver diseases. NASH was defined pathologically using the modified Matteoni’s classification, as shown in the study of Rafiq et al. [31]. Subtypes 3 and 4 were considered as NASH in this study. Tumor depth was defined as the distance from the hepatic surface to the nearest tumor surface upon preoperative diagnostic imaging. The liver anatomy and operative methods were classified according to the Brisbane 2000 terminology [32].

## 3. Results

### 3.1. Surgical and Pathological Outcomes of Patients Who Underwent RAH

Table 1 shows the clinical characteristics and surgical outcomes of the patients. Five patients presented with hepatitis B virus infection, nine with hepatitis C virus infection, four with alcoholic liver disease, and four with nonalcoholic steatohepatitis. In 30 patients with 33 collective tumors who underwent RAH, the tumor locations were at S2 (*n* = 7), S3 (*n* = 7), S4 (*n* = 4), S5 (*n* = 3), S6 (*n* = 5), S7 (*n* = 3), and S8 (*n* = 4). In total, 28 and 2 patients underwent partial hepatic resection and anatomic resection (lateral sectionectomy and S7 segmentectomy), respectively. In patient no. 30 who underwent lateral sectionectomy, ICG was injected intravenously after hepatic parenchymal dissection to check the blood flow in the remnant liver. In patient no. 24 who underwent S7 segmentectomy, intraoperative ICG infusion into the portal branch of S6 was performed to identify the boundary between S6 and S7 after ICG fluorescence tumor identification. The median tumor diameter and tumor depth were 2.1 (range: 0.5–5.1) and 0 (0–1.8) cm, respectively. The median operative time and volume of blood loss were 259 (124–594) min and 150 (1–1350) mL, respectively. One patient presented with chronic heart failure, which was a Clavien–Dindo grade IIIa postoperative complication [33]. None of the patients developed other complications including bile leakage caused by surgical procedures. Further, 20 tumors were pathologically diagnosed as hepatocellular carcinoma (HCC), 2 as intrahepatic cholangiocarcinoma (ICC), 1 as combined HCC, 9 as metastatic liver tumor, and 1 as hemangioma.

Further, 28 patients with 31 tumors underwent ICG fluorescence imaging during RAH. Preoperative intravenous ICG injection was performed 2–6 (median: 2) days before surgery. In total, 26 (84%) tumors were identified on hepatic surfaces via fluorescence imaging prior to hepatic transection (Figure 1, Appendix A). All five tumors with tissue or peritumoral fluorescence signals that could not be identified via ICG fluorescence imaging were located at a depth of >0.8 cm from the hepatic surface. The fluorescence signals of eight tumors were detected from the hepatic raw surfaces during parenchymal dissection, enabling surgeons to adjust the transection planes to ensure appropriate surgical margins (Figure 2, Appendix A). One patient with intrahepatic cholangiocarcinoma (patient no. 25) tested positive for cancer cells in the dissected stump of the bile duct. However, pathological examinations confirmed negative surgical margins surrounding the dissected hepatic parenchyma via ICG fluorescence imaging in all tumors. On the contrary, one (patient no. 15) of two patients with ICG contraindications had a positive surgical margin at the site of the dissected hepatic parenchyma.

### 3.2. Patient No. 15

Patient no. 15 could not undergo ICG fluorescence imaging because of an iodine allergy. Therefore, they only underwent RAH with normal color imaging. Partial hepatectomy at S4 was performed, and tumor exposure at the hepatic raw surface was suspected during clamp-crushing hepatic parenchymal dissection (Figure 3A). Pathological examination revealed invasion of the dissected hepatic parenchyma in ICC (Figure 3B). 

### 3.3. Patient No. 25

Patient no. 25 underwent partial hepatectomy at S4. The tumor was detected at the hepatic surface prior to parenchymal dissection via ICG fluorescence imaging (Figure 4A). In addition, the peritumoral fluorescence signal was observed at the hepatic raw surface during hepatic parenchymal dissection (Figure 4B), and the transection plane was adjusted to prevent tumor exposure at the hepatic raw surface. The feasible surgical margin was achieved surrounding the dissected hepatic parenchyma. However, pathological examination revealed invasion of the stump of the dissected bile duct in ICC (Figure 4C).

## 4. Discussion

The current study showed that ICG fluorescence imaging detected 26 (84%) of 31 tumors on the hepatic surface prior to hepatic transection. In addition, one tumor had a positive surgical margin because of microscopic biliary infiltration. However, pathological tumor invasion of the dissected hepatic parenchyma was not observed in patients who underwent ICG fluorescence imaging in RAH. By contrast, in one of two patients who could not undergo ICG fluorescence imaging, pathological tumor invasion of the dissected hepatic parenchyma was observed.

ICG fluorescence imaging is based on the emission of fluorescence signals by protein-bound ICG when illuminated by near-infrared light between 710 and 810 nm [9]. After intravenous injection, the ICG is taken up into the hepatocytes by membrane transporters (organic anion-transporting polypeptide-8, Na^+^/taurocholate co-transporting polypeptide), and almost 100% of the ICG is excreted in the bile juice [34]. In well-to-moderately differentiated HCC, the membrane transporters involved in ICG uptake remain on the cancer cell surface. Nevertheless, the biliary excretion process is impaired. Thus, well-to-moderately differentiated HCC is fluorescently labeled [6]. By contrast, such membrane transporters are lost in poorly differentiated HCC and adenocarcinomas. Therefore, a rim-type fluorescent signal is shown, reflecting bile stasis in the surrounding liver parenchyma caused by the tumor [6]. Tumor or peritumoral fluorescent signals remain for a long period. Thus, fluorescent signals can be confirmed intraoperatively if ICG is administered intravenously within 14 days before surgery for well-differentiated and moderately differentiated HCC and within several days before surgery for poorly differentiated HCC and adenocarcinomas [6,29]. However, the fluorescent signal of the tumor or the surrounding area is difficult to confirm when ICG is administered the day before or on the day of surgery due to fluorescence of the entire liver parenchyma. Therefore, preoperative intravenous ICG was injected 2–6 (median: 2) days before surgery in this study. A previous study reported that the tumor detectability rate during laparoscopic hepatectomy was 85% [35], and it was similar to that in the current study (84%). Tumor detectability is strongly influenced by tumor depth [6,35]. In this study, all five tumors that could not be detected fluorescently were located at a depth of approximately >0.8 cm from the hepatic surface. Thus, identifying tumors using ICG fluorescence imaging is not possible in all cases. Tumors should be identified using conventional imaging modalities, such as preoperative contrast-enhanced computed tomography, magnetic resonance imaging, or intraoperative ultrasound sonography, that can detect tumors at a depth of more than 0.5–1 cm [36]. ICG fluorescence imaging should only be used as an adjunctive modality. However, combining ICG fluorescence imaging with conventional modalities may facilitate clearer and easier tumor identification, leading to more curative treatments. In fact, a recent meta-analysis revealed that ICG fluorescence-guided open or laparoscopic hepatectomy improved the R0 resection rate [16]. Despite its limitations, ICG fluorescence imaging is an easy and useful technique for tumor identification.

The tumor or peritumoral fluorescence signals cannot be detected in tumors located at a depth of >0.8 cm from the hepatic surface [6,35], as described above. This also indicates that fluorescence signals in the dissected raw surface can be assessed to achieve appropriate surgical margins at a depth of approximately >0.8 cm. Actually, previous research evaluated 14 patients who underwent RAH for colorectal liver metastases. The results showed that all patients who tested positive for cancer cells at the site of the dissected surface pathologically presented with fluorescence signals at the dissected hepatic raw surface in the resected specimen [29]. In the current study, eight patients required transection plane adjustment because the fluorescence signals were detected at the hepatic raw surfaces during parenchymal dissection. Nevertheless, pathological tumor invasion of the dissected hepatic parenchyma could be prevented with ICG fluorescence imaging in all patients. In RAH, in which the sense of touch is completely lost, the hepatic transection planes should be adjusted appropriately with reference to fluorescence signals to achieve negative surgical margins surrounding the dissected hepatic parenchyma. 

One patient with ICC who underwent fluorescence-guided RAH tested positive for cancer cells in the stump of the dissected bile duct. In this patient, a positive surgical margin at the hepatic raw surface could be prevented by adjusting the transection plane with reference to the fluorescence signal. However, although it was far from the tumor body macroscopically, microscopic invasion into the stump of the bile duct was observed. Tumor detection via ICG fluorescence imaging is simply a technique used to visualize cholestasis in or around tumors, not specifically for the fluorescent labeling of tumor cells [6]. In the current case, if biliary obstruction caused by macroscopic tumor invasion was observed, fluorescence signals attributed to cholestasis might help achieve appropriate surgical margins. However, microscopic tumor invasion cannot be detected via ICG fluorescence imaging, and this is also one of the limitations of ICG fluorescence imaging in tumor visualization. By contrast, a previous study using in vitro patient samples reported the use of a novel fluorescent probe that is hydrolyzed with γ-glutamyl transpeptidase, which is overexpressed in adenocarcinoma tissues. The results showed that the probe rapidly detected fluorescence in the visible region [37,38]. This may fluorescently label ICC and colorectal liver metastasis, and the risk of recurrence after resection can be predicted by measuring the intensity of the fluorescence [38]. If cancer-specific fluorescence visualization is established clinically, the usefulness of fluorescence-guided surgery will be more important.

In this study, one of the two patients who could not undergo ICG fluorescence imaging presented with pathological tumor invasion of the dissected hepatic parenchyma. The tumor was exposed at the hepatic raw surface during hepatic parenchymal dissection and was diagnosed pathologically as ICC. In case of open or laparoscopic hepatectomy, the induration might be palpated during dissection around the adenocarcinoma. However, the sense of touch is completely lost in RAH, as described above, and this might have been the reason why the tumor was exposed, as the induration could not be palpated in the dissection around the tumor. This case suggested that it was occasionally challenging to achieve appropriate surgical margins in RAH. Some studies have reported the indications of laparoscopic hepatectomy in ICC [39,40,41], and achieving a sufficient margin is an important factor in determining the indication of laparoscopic hepatectomy in ICC. Furthermore, some patients presented with tumor invasion in the stump of the bile duct even without evident tumor exposure, as in one patient presented earlier. The use of ICG fluorescence imaging may be useful for securing margins in the dissected raw surface. However, at present, these characteristics require cautious indication for RAH in ICC.

The current study had some limitations. First, this retrospective study included a small number of patients at a single institution. Second, an accurate assessment might not have been performed, considering that several cases were in the introductory phase at our institution and that the learning curve in RAH requires an appropriate number of cases [42,43]. By contrast, although RAH is mainly implemented only at high-volume centers, it can be developed further in the near future. In eight of the cases utilizing ICG fluorescence imaging in this study, the data from the RAH introductory phase showed that the transection plane was modified based on the fluorescence signal, and tumor exposure to dissected hepatic parenchyma was avoided. Appropriate surgical margins are sometimes technically difficult to achieve, especially during the introductory phase of RAH, and ICG fluorescence tumor identification is a useful tool that can complement the surgeon’s experience in this regard. ICG fluorescence imaging, which is recommended based on surgical outcomes during the introduction phase at our institution, can play an important role in the spread of RAH.

## 5. Conclusions

ICG fluorescence imaging facilitates the easy identification of hepatic tumors even in RAH. Hence, it can be helpful for achieving appropriate surgical margins at the hepatic dissected raw surface.

## Figures and Tables

**Figure 1 cancers-15-04205-f001:**
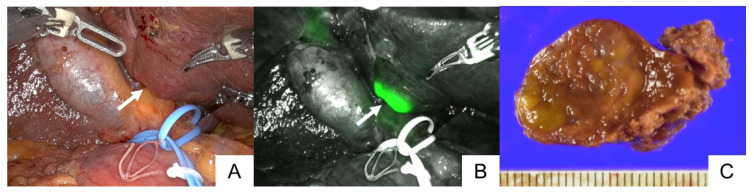
Tumor identification via ICG fluorescence imaging. The tumor at the hepatic surface ((**A**), arrow) was identified based on fluorescence signals with near-infrared light ((**B**), arrow). The tumor was diagnosed as hepatocellular carcinoma (**C**).

**Figure 2 cancers-15-04205-f002:**
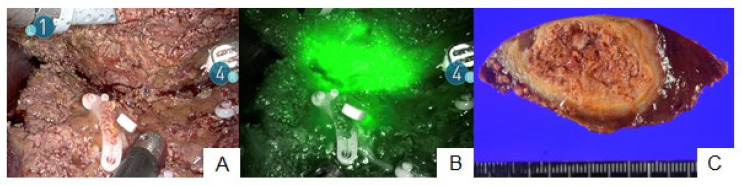
Fluorescence signals at the hepatic raw surface. Although normal color imaging showed the dissected hepatic raw surface (**A**), ICG fluorescence imaging revealed an insufficient surgical margin with the fluorescence signal of the tumor at the hepatic raw surface (**B**). A pathological negative surgical margin was achieved (**C**).

**Figure 3 cancers-15-04205-f003:**
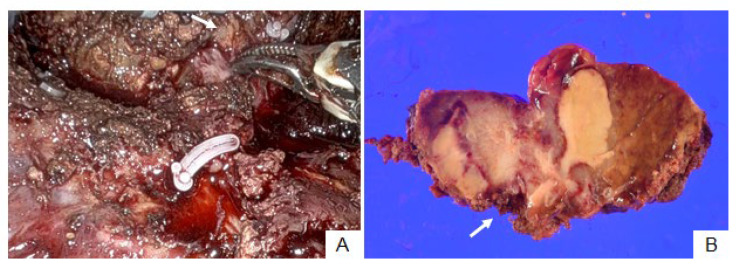
Patient no. 15. Intrahepatic cholangiocarcinoma was exposed at the dissected hepatic raw surface during hepatic parenchymal dissection ((**A**), arrow). Cancer cells invaded the dissected hepatic parenchyma ((**B**), arrow).

**Figure 4 cancers-15-04205-f004:**
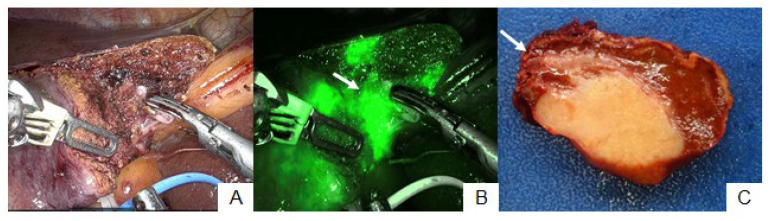
Patient no. 25. The hepatic transection plane was adjusted (**A**) because the peritumoral fluorescence signal was detected during hepatic parenchymal dissection ((**B**), arrow). The feasible surgical margin was achieved around the dissected hepatic parenchyma (**C**). However, cancer cells invaded the stump of the dissected bile duct microscopically ((**C**), arrow).

**Table 1 cancers-15-04205-t001:** Clinical characteristics and surgical outcomes of patients who underwent robot-assisted hepatectomy.

Patient No.	Presence of Liver Disease	Tumor Number	Tumor Location	Tumor Diameter (cm)	Tumor Depth (cm) *	Pathological Diagnosis	ICG Fluorescence Imaging	Fluorescence Tumor Identification at the Hepatic Surface	Fluorescence Signal at the Hepatic Raw Surface	Surgical Margin
1	None	1	S5	1.0	0	MLT	Yes	Yes	No	Negative
2	ALD	1	S6	5.1	0	HCC	Yes	Yes	No	Negative
3	None	1	S8	1.4	1.8	MLT	Yes	No	No	Negative
4	HCV	1	S3	2.8	0	HCC	Yes	Yes	No	Negative
5	HBV	1	S8	1.4	0	Combined HCC	Yes	Yes	Yes	Negative
6	None	1	S5	3.1	0	ICC	Yes	Yes	Yes	Negative
7	ALD	1	S3	2.2	0	HCC	Yes	Yes	No	Negative
8	HCV	1	S2	2.5	0	HCC	Yes	Yes	No	Negative
9	HCV	1	S6	1.6	0	HCC	Yes	Yes	No	Negative
10	HCV	1	S5	2.0	0	HCC	Yes	Yes	No	Negative
11	ALD	1	S7	2.2	0	HCC	Yes	Yes	No	Negative
12	HCV	1	S2	3.3	0	HCC	Yes	Yes	No	Negative
13	HCV	1	S6	1.0	0	HCC	No	-	-	Negative
14	HCV	1	S4	1.3	0.5	HCC	Yes	Yes	No	Negative
15	NASH	1	S4	4.1	0	HCC	No	-	-	Positive (parenchyma)
16	HCV	1	S7	2.1	0	HCC	Yes	Yes	No	Negative
17	NASH	1	S2	4.9	0	HCC	Yes	Yes	No	Negative
18	None	1	S2	2.2	0	MLT	Yes	Yes	Yes	Negative
19	NASH	1	S3	2.9	0	HCC	Yes	Yes	Yes	Negative
20	None	1	S2	2.1	0.8	MLT	Yes	No	No	Negative
21	None	1	S3	1.8	0.2	HCC	Yes	Yes	No	Negative
22	HBV	1	S8	3.5	0	HCC	Yes	Yes	Yes	Negative
23	None	3	S2	0.6	0	MLT	Yes	Yes	No	Negative
			S3	0.5	0			Yes	No	Negative
			S4	2.4	1.2			No	Yes	Negative
24	ALD	1	S7	3.0	1.3	HCC	Yes	No	Yes	Negative
25	HCV	1	S4	3.5	0	ICC	Yes	Yes	Yes	Positive (bile duct)
26	NASH	1	S3	2.0	0	HCC	Yes	Yes	No	Negative
27	None	2	S3	0.8	0.3	MLT	Yes	Yes	No	Negative
			S8	0.8	0.8	Hemangioma		No	No	Negative
28	HBV	1	S6	2.1	0	HCC	Yes	Yes	No	Negative
29	HBV	1	S6	1.0	0	MLT	Yes	Yes	No	Negative
30	HBV	1	S2	4.0	0	ICC	Yes	Yes	No	Negative

ALD—Alcoholic liver disease; HCV—hepatitis C virus; HBV—hepatitis B virus; NASH—nonalcoholic steatohepatitis; MLT—metastatic liver tumor; HCC—hepatocellular carcinoma; ICC—intrahepatic cholangiocarcinoma; ICG—indocyanine green. * Tumor depth was defined as the distance from the hepatic surface to the nearest tumor surface.

## Data Availability

All data generated or analyzed during this study are included in this article. Further enquiries can be directed to the corresponding author.

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
