# Peer review of "Application of Indocyanine Green Fluorescence Imaging for Tumor Localization during Robot-Assisted Hepatectomy"

_cancers, 2023, doi:10.3390/cancers15174205_

Round 1

Reviewer 1 Report

This is an interesting surgical series of RAH at a single institution discussing the applicability of indocyanine green fluorescence imaging for tumor localization during RAH.  There are a few minor suggested corrections in content and grammar listed below:

Lines 55-56 refer to robotic surgery in other fields than "robotic assisted hepatectomy".  The listed references  are for nephrectomy, metabolic and bariatric surgery, and colectomy - not RAH.  Rewrite this sentence. Perhaps "Robotic assisted surgery".

Line 63 perhaps change "may cover for" to  "could compensate for"

Line 205 change "cannot" to "could not"

Line 210 change "if" to "when"

Line 229 "at depths approximately >0.8CM" rather than "at a depth of approximately > 0.8 cm.

The language is fine except for instance shown above.

Author Response

We thank the reviewer for these pertinent comments. The suggested points were revised as requested. (Lines 56, 65, 213-214, 217, 237-238)

Reviewer 2 Report

Kinoshita et al applied indocyanine green fluorescence to visualize hepatic tumors in robot-assisted hepatectomy. Although ICG has proved to be very useful in visualization of hepatic tumors by previous research, this work present good supplement to hepatectomy. The data in the manuscript support the conclusion very well and should be useful for further clinical research. My recommendation is to revise the paper faintly for the reasons specified below

1.      For 2 patients underwent anatomic resection, why not been injected with ICG during the operation?

2.      What is the advantage of RAH combined with ICG imaging compared to the open hepatectomy, especially? please explain.

Minor editing of English language required.

Author Response

We thank for your insightful review.

Comment 1

For 2 patients underwent anatomic resection, why not been injected with ICG during the operation?

Response to comment 1

We thank the reviewer for this comment. In 2 anatomic resection cases, ICG was injected intraoperatively to perform negative or positive staining after ICG fluorescence tumor identification. This should be described in manuscript. Therefore, we added sentences in Results section as below;

“In patients who underwent lateral sectionectomy, ICG was injected intravenously after hepatic parenchymal dissection to check the blood flow in the remnant liver. In patients who underwent S7 segmentectomy, intraoperative ICG infusion into portal branch of S6 was performed to identify the boundary between S6 and S7 after ICG fluorescence tu-mor identification.” (Lines 140-144)

Comment 2

What is the advantage of RAH combined with ICG imaging compared to the open hepatectomy, especially? Please explain.

Response to comment 2

We thank the reviewer for this comment. The previous study reported that ICG fluorescence tumor identification has been shown to be more useful in laparoscopic hepatectomy than in open hepatectomy because of poorer palpation and easier to use. Similarly, ICG fluorescence imaging may be useful in RAH as a tool that can compensate for the lack of palpation, but there have been few reports of this technique, which led to the inspiration for this study. This should be described clearly in text. Therefore, we added sentences in Introduction section as below;

“In particular, due to poorer palpation and easier to use, the visualization of hepatic tumors is a more efficient technique in laparoscopic hepatectomy than in open hepa-tectomy [18].” (Lines 53-55)

“and this may be highly useful tool as well as laparoscopic hepatectomy.” (Lines 65-66)

Reviewer 3 Report

In this retrospective study, the authors comprehensively summarized and analyzed the efficacy and usefulness of indocyanine green (ICG) fluorescence imaging for visualizing hepatic tumors in robot-assisted hepatectomy in their institution. It revealed that the application of ICG could obviously lighten and identify the tumor on the hepatic surface (84%) and provide strong evidence to enable surgeons to adjust the transection planes to ensure appropriate surgical margins of tumor located on hepatic surfaces prior to hepatic transection. Significantly, the application of ICG realized totally negative surgical margins at the dissected hepatic parenchyma. Overall, ICG fluorescence imaging could facilitate the easy identification of hepatic tumors even in RAH and would be useful for confirming appropriate surgical margins. This retrospective study is well organized in the summarization of the application of ICG imaging for tumor localization during robot-assisted hepatectomy, and especially and deeply analyzed the potential application, advantages, limitations, and reason analysis. They also openly present some limitations of this study. Therefore, there is no need for more revision for this manuscript to publish on Cancers.

 Minor editing of English language required

Author Response

Thanks for your kind comment.